# Assessment of Infection Prevention and Control Measures at Points of Entry in Sierra Leone in 2021: A Cross-Sectional Study

**DOI:** 10.3390/ijerph19105936

**Published:** 2022-05-13

**Authors:** Kadijatu Nabie Kamara, James Sylvester Squire, Joseph Sam Kanu, Ronald Carshon-Marsh, Zikan Koroma, Aminata Tigiedankay Koroma, Anna Maruta, Christiana Kallon, Marcel Manzi, Bienvenu Salim Camara, Aelita Sargsyan, Alexandre Delamou, Jamie Ann Guth, Anthony Reid, Mohamed Ahmed Khogali, Mohamed Alex Vandi

**Affiliations:** 1National Disease Surveillance Programme, Directorate of Health Security and Emergencies (DHSE), Ministry of Health and Sanitation (MoHS), Cockerill, Wilkinson Road, Freetown 00232, Sierra Leone; jmssquire@yahoo.com (J.S.S.); samjokanu@yahoo.com (J.S.K.); aminata_krm@yahoo.com (A.T.K.); 2National Malaria Control Progamme, MoHS, Freetown 00232, Sierra Leone; naldoline@yahoo.com; 3Clinical Laboratories, Directorate of Laboratory and Blood Services, MoHS, Freetown 00232, Sierra Leone; zikankoroma@gmail.com; 4IPC/AMR Team Lead, World Health Organization (WHO) Country Office, 21A/B Riverside Drive, Freetown 00232, Sierra Leone; marutaa@who.int; 5Infection Prevention and Control Unit, DHSE, MoHS, Freetown 00232, Sierra Leone; christy.conteh@yahoo.com; 6Independent Researcher, 5000 Namur, Belgium; m.manzi449@gmail.com; 7Maferinyah National Research and Training Centre, Ministry of Health, Conakry BP 1147, Guinea; bienvenusalimcamara@gmail.com; 8Tuberculosis Research and Prevention Centre, 6/2 Adonts Str., 100 Apt., Yerevan 0014, Armenia; sargsyan.aelita@gmail.com; 9National Training and Research Centre in Rural Health, 01 Maferinyah, Forecariah BP 2649, Guinea; adelamou@maferinyah.org; 10Global Health Connections, Center Barnstead, Barnstead, NH 03225, USA; guth.jamie@gmail.com; 11LuxOR, Operational Research Unit, Médecins Sans Frontières, 68 rue Gasperich, L-1617 Luxembourg, Luxembourg; tony.reid@brussels.msf.org; 12Special Programme for Research and Training in Tropical Diseases (TDR), 1211 Geneva, Switzerland; khogalim@who.int; 13DHSE, MoHS, Cockerill, Wilkinson Road, Freetown 00232, Sierra Leone; mohamedavandi69@yahoo.com

**Keywords:** infection prevention and control, points of entry, SORT IT (Structured Operational Research Training Initiative), Infection Prevention Control Assessment Framework (IPCAF), antimicrobial resistance, IPC at PoEs, Sierra Leone IPC, Sierra Leone PoEs

## Abstract

Implementing and monitoring infection prevention and control (IPC) measures at immigration points of entry (PoEs) is key to preventing infections, reducing excessive use of antimicrobials, and tackling antimicrobial resistance (AMR). Sierra Leone has been implementing IPC measures at four PoEs (Queen Elizabeth II Quay port, Lungi International Airport, and the Jendema and Gbalamuya ground crossings) since the last Ebola outbreak in 2014–2015. We adapted the World Health Organization IPC Assessment Framework tool to assess these measures and identify any gaps in their components at each PoE through a cross-sectional study in May 2021. IPC measures were Inadequate (0–25%) at Queen Elizabeth II Quay port (21%; 11/53) and Jendema (25%; 13/53) and Basic (26–50%) at Lungi International Airport (40%; 21/53) and Gbalamuya (49%; 26/53). IPC components with the highest scores were: having a referral system (85%; 17/20), cleaning and sanitation (63%; 15/24), and having a screening station (59%; 19/32). The lowest scores (0% each) were reported for the availability of IPC guidelines and monitoring of IPC practices. This was the first study in Sierra Leone highlighting significant gaps in the implementation of IPC measures at PoEs. We call on the AMR multisectoral coordinating committee to enhance IPC measures at all PoEs.

## 1. Introduction

Antimicrobial resistance (AMR), defined as the ability of microbes to resist the effect of antimicrobials, is a significant public health problem globally [1]. To date, AMR has been responsible for 700,000 deaths per year across the world, and this figure is predicted to reach 10 million per year by 2050 [2]. In addition, AMR has economic and societal implications. These include increased health costs because of the longer duration of treatment and the need for additional tests and more expensive alternative drugs [3,4,5,6]. The burden of AMR is higher in low- and middle-income countries (LMICs) than in high-income countries because of the higher burden of infectious diseases; lack of infrastructure such as well-equipped laboratories, clean water, and sanitation; and limited human and financial resources to adequately address AMR [7].

The main reason for the development of AMR is the excessive and inappropriate use of antimicrobials [1,2,8]. This implies that a reduction in antimicrobial use will potentially reduce resistance [8]. One of the most effective ways to reduce the overuse of antimicrobials is to prevent the occurrence of infection in the first place [9]. In 2015, the World Health Organization (WHO) developed the Global Action Plan (GAP) to tackle AMR [10]. One of the main pillars of the GAP is to reduce the incidence of infection through effective sanitation, hygiene, and infection prevention measures as “every infection prevented is one antibiotic treatment avoided” [10].

One of the main sites, in addition to health facilities, where infection prevention and control (IPC) measures need to be implemented and monitored to prevent the spread of infectious diseases as a result of population movement are immigration points of entry (PoEs) [11,12]. As people cross borders, they may cross with infectious diseases including those caused by antimicrobial-r strains [13]. PoEs act as “frontline” facilities dealing with potentially many exposures to infectious diseases. A PoE is defined as a “passage for international entry or exit of travelers, baggage, cargo, containers, conveyances, goods, and postal parcels, as well as agencies and areas providing services to them on entry or exit”. They include airports, seaports, and ground crossings [14]. Implementing and monitoring IPC measures at PoEs has become more important than ever before because of the global emergence and spread of the novel coronavirus disease (COVID-19) [11].

Sierra Leone is a country in West Africa that has been affected by Ebola virus disease (EVD), Lassa fever, and cholera as well as the current COVID-19 pandemic. One of the main reasons for the emergence and spread of these diseases is population movement between Sierra Leone and neighboring countries [15,16].

Due to ongoing exposure to infectious diseases, the government of Sierra Leone implemented IPC measures in 2014 during the 2014–2015 EVD outbreak at the four designated class A PoEs for travelers. Of equal importance to implementation is the monitoring and assessment of these measures. While WHO has developed the IPC Assessment Framework (IPCAF) tool for health facilities [17], as of May 2021, there was no specific tool dedicated to assessing IPC measures at PoEs.

To address this gap, we adapted the IPCAF tool to assess the IPC measures at four designated class A (high flow) PoEs. It is important to determine the quality of IPC at PoEs, identify gaps, and formulate recommendations for decision-makers to improve IPC. The information should also provide baseline data for future monitoring and planning. To date, no such assessment has been conducted in the country.

We carried out this study to: (a) assess current IPC measures at four PoEs in Sierra Leone using an adapted WHO IPCAF tool, (b) identify gaps in components of IPC measures for each PoE, and (c) formulate recommendations for decision-makers to improve IPC.

## 2. Materials and Methods

### 2.1. Study Design

This was a cross-sectional study.

### 2.2. General Setting

Sierra Leone is a country in West Africa bordered by Guinea to the northeast and Liberia to the southeast. It has a total surface area of 71,740 km^2^ and an estimated population of about eight million [18].

Sierra Leone’s economy has been affected by 11 years of civil war (1991–2002) [19] and by the EVD outbreak in 2014–2015 [16,20]. The country has a weak health system that is burdened with several infectious diseases. It also has a history of cross-border infectious disease spread, including the 2014–2015 EVD outbreak, the 2018–2019 measles outbreak in the Kambia district [15,16], and now COVID-19. In most instances, the same ethnic groups live on both sides of the borders between Sierra Leone and its neighboring countries, Liberia and Guinea. These interconnections along the borders increase the risk of cross-border spread of infectious diseases including the spread of AMR.

Sierra Leone has a total of 346 PoEs in 13 of its 16 districts. The majority (91%; 316/346) of these PoEs are porous crossing points as they are not manned. The PoEs are tiered into class A (high flow), class B (medium flow), and class C (low flow). There are four designated class A PoEs in the country.

### 2.3. Specific Setting

This study was conducted at the four designated class A PoEs of Sierra Leone, namely, Lungi International Airport, Queen Elizabeth II Quay port, and the Gbalamuya and Jendema ground crossing points (Appendix A). Lungi International Airport is situated in Port Loko district, and Queen Elizabeth II Quay is in Freetown, the capital city of Sierra Leone. The Gbalamuya and Jendema ground crossing points are located in Kambia district, bordering Guinea, and in Pujehun district, bordering Liberia (Figure 1), respectively. All four PoEs have a full complement of staff/officers from customs, immigration, agriculture, the Office of National Security, the Pharmacy Board of Sierra Leone, security forces (police and soldiers), port health, cleaners, and other support staff. The port health staff are either public health or clinical officers responsible for screening travelers for signs and symptoms of infectious diseases, checking for vaccination status of travelers, and ensuring environmental sanitation. They also monitor and evaluate all foodstuffs, cosmetics, disinfectants, hazardous substances, and medicines entering or exiting the country. The staff strength of the four PoEs includes a total of 76 staff: Lungi International Airport, 26; Queen Elizabeth II Quay, 17; Gbalamuya, 20; and Jendema, 11. The participation rate at Lungi International Airport was 46% (12/26), Queen Elizabeth II Quay 65% (11/17), Gbalamuya 65% (13/20), and Jendema 100% (11/11), making an overall participation rate of 62% (47/76).

### 2.4. The National Port Health Units

The International Health Regulation (IHR), an instrument of international law that legally binds the WHO member states, requires states to establish and maintain core capacities including at designated PoEs [14]. IHR recommendations and the 2014–2015 EVD outbreak in the country accelerated efforts to build IPC capacity at PoEs in Sierra Leone. These included strengthening port health units and forming a national coordinating body that sits at the Sierra Leone National Public Health Emergency Operations Centre, Directorate of Health Security and Emergencies, Ministry of Health and Sanitation. The national port health unit provides leadership and coordinates and monitors the implementation of the core capacities at the PoEs as recommended by the IHR. These capacities include access to medical services, transport of ill travelers, inspection of conveyances, arrangements for isolation (human, animal), and control of vectors/reservoirs. Also, in order to respond to events, they should have emergency contingency plans, arrangements for isolation, space for interviews/quarantines, and the capacity to apply specific control measures [14].

### 2.5. Study Population and Period

This study involved interviewing port health staff at all four PoEs and observing IPC measures during May 2021.

### 2.6. Adapted WHO IPC Checklist and Variables

This study was conducted using a tool (Appendix A) adapted from a standardized IPCAF tool (2018 version) [17]. It is a systematic tool that can provide a baseline assessment of the IPC program and activities within a health care facility as well as ongoing evaluations through repeated administration to document progress over time and facilitate improvement. It is a structured, close-ended questionnaire with a scoring system on eight components. Based on the overall percentage achieved in the eight sections, a facility is assigned to one of four levels of IPC promotion and practice (Table 1).

Adaptation of the tool was performed in consultation with the national IPC unit, the port health unit, and the WHO Country Office IPC/AMR team. Several sections of the original tool were removed as they were not relevant to the PoE setting. These included: Core component 1 (IPC program); Core component 4 (health care-associated infection surveillance); Core component 5 (multimodal strategies for implementation of IPC interventions); and Core component 7 (workload, staffing, and bed occupancy). The following four sections were kept in the adapted tool: Core component 2 (IPC guidelines); Core component 3: (IPC education and training); Core component 6 (monitoring/audit of IPC practices and feedback); and some aspects of Core component 8 (built environment, materials, and equipment for IPC at the facility level) that we named cleaning and sanitation. Three other sections (screening station, isolation facility, and referral system) were developed and added to the tool based on the IHR recommendation for core capacities at PoEs including screening, isolation, and a referral system for sick/suspected travelers.

Each section of the adapted tool has subcomponent questions with a total of 53 questions for the assessment. Most questions have a yes or no response and are coded as 1 or 0, respectively. On some questions, compliance was graded and scored as: none (0), partial (0.5), or full (1). The total number of responses was added and divided by the total number of questions for that section. This was multiplied by 100 to obtain the percentage scores. Based on the overall percentage scores in the seven sections, each POE was rated at one of four levels of IPC promotion and practice as indicated in the WHO IPCAF tool: Inadequate (0–25%), Basic (26–50%), Intermediate (51–75%), or Advanced (76–100%).

### 2.7. Data Sources, Data Variables, Data Validation, and Data Analysis

Data variables included date, district, PoE name, PoE type, PoE class, and IPC component scores. Data were collected by the principal investigator (PI) together with a national port health staff member and IPC staff. The checklist was administered to port health staff by face-to-face interviews after pretesting. The data were entered into a Microsoft Excel spreadsheet at the national level by dedicated data entry clerks and validated by the PI through a random sample of 10% of the assessment checklist. These data were compared with those entered in Microsoft Excel. Where there were errors, further elaborate cross-checking was done.

The data in Microsoft Excel were analyzed using descriptive statistics and results expressed as frequencies and percentages. Average scores for each IPC component were computed for the four PoEs. To identify specific gaps, we listed the subcomponents in each thematic area with zero scores for IPC.

## 3. Results

To assess current IPC measures at the four PoEs, we interviewed 47 port health staff members: 13 at Gbalamuya ground crossing point, 12 at Lungi International Airport, and 11 each at Queen Elizabeth II Quay seaport and Jendema ground crossing point.

### 3.1. Availability and Score of Each IPC Component at Individual POE

Table 2 shows the scores of IPC measures for the individual PoEs. At all four PoEs, no guidelines or standard operating procedures (SOP) on IPC activities were available, and there was no monitoring of IPC practices.

Concerning IPC training, only two (50%) of the PoEs reported having received basic orientation on IPC.

The availability of infrastructure and materials for screening stations varied from 13% (1/8) at Queen Elizabeth II Quay to 88% (7/8) at Gbalamuya ground crossing.

Materials for and frequency of environmental cleaning and sanitation were lowest at Jendema ground crossing (17%; 1/6) and highest at Lungi International Airport (100%; 6/6).

At all four PoEs, some system was in place for referral of sick or suspected travelers.

### 3.2. Score and Level of Each IPC Component Measure at the Four PoEs Collectively

Table 3 summarizes scores on IPC measures at the four class A PoEs in Sierra Leone. IPC measures varied substantially by component. Referral system had the highest reported score (85%; 17/20) followed by cleaning and sanitation (63%; 15/24), with IPC guidelines and monitoring of IPC practices scoring 0% each.

### 3.3. Identified Gaps in Specific IPC Components at the Four PoEs

Detailed, specific gaps in IPC at the four PoEs are presented in Table 4. Guidelines and periodic monitoring of IPC compliance were absent in all PoEs. They also all lacked isolation areas for suspected/sick travelers and personal protective equipment (PPE).

Three PoEs (Lungi International Airport, Queen Elizabeth II Quay, and Gbalamuya ground crossing point) reported having a structure for isolation of travelers, but only one (Gbalamuya) had a permanent structure (with no toilet facility), while the other two PoEs (Lungi International Airport and Queen Elizabeth II Quay) had temporary tents that were set up.

## 4. Discussion

This was the first study in Sierra Leone to assess the implementation of IPC measures at the four class A PoEs. This assessment is critical as frontline PoE staff are regularly exposed to travelers and goods that may carry infectious diseases. The findings of this study could serve as a baseline reference for future assessments to monitor IPC compliance. Good IPC protects PoE staff as well as travelers passing through the PoE [11]. The WHO IPCAF tool that was adapted for this study in consultation with the WHO Country Office IPC/AMR team, national IPC, and port health units of the Ministry of Health and Sanitation was shown to be feasible.

Overall, our study showed that the level of implementation of IPC measures was Inadequate in two of the four PoEs (Queen Elizabeth II Quay port and Jendema ground crossing point) and Basic at Lungi International Airport and Gbalamuya ground crossing point. The relatively higher scores at the latter two could be explained by the fact that these PoEs were prioritized during the COVID-19 pandemic due to higher numbers of travelers passing through them compared with the others. The low level, overall, of IPC measures at PoEs observed in our study can be explained by the fact that resources are mostly allocated to health facilities. Given the large exposure to travelers, who may harbor infectious diseases, port facilities and staff are vulnerable to infections that may also become resistant to antimicrobials.

Discussion of AMR prevention through IPC has focused on preventing health care-associated infections (HAIs) in health facilities [9], where the most difficult-to-treat antibiotic-resistant infections occur. The recommendations for implementing and strengthening IPC measures made by the WHO GAP focused on health facilities [10]. However, establishing or improving IPC measures at PoEs can help prevent cross-border infection, thereby reducing antimicrobial use and possibly AMR occurrence.

Instituting effective IPC measures at PoEs is critical for minimizing or preventing infection spread, including the prevention of AMR. The movement of patients across borders has been identified as a risk factor for the introduction of carbapenemase-producing Enterobacteriaceae into health care settings and systems [21]. A systematic review by Mouchtouri and colleagues on exit and entry screening practices for infectious diseases among travelers at PoEs concluded that exit screening measures in affected areas were very important and should be applied together with other preventive measures including epidemiological investigation, information strategies, vaccination, quarantine, and contact tracing to achieve comprehensive containment of disease outbreaks [22]. At PoEs, management of high volumes of exposed or infected travelers can be very challenging, and this has the potential to negatively impact trade and the economy. In the current COVID-19 pandemic and during the severe acute respiratory distress syndrome (SARS) outbreak in 2002–2004, for instance, there was the global implementation of entry and exit screening and contact tracing at PoEs worldwide [23,24,25].

To ensure global health security, there is a need for countries to develop robust multisectoral systems to rapidly detect and respond to both imported and domestic communicable diseases [26]. This is critical as recent multinational disease outbreaks have demonstrated the risk of disease spreading globally before public health systems can respond to an event [24].

One of the strengths of this study is that we were able to assess all four main designated immigration PoEs, thus providing a national baseline report. Second, the adaptation of the WHO IPCAF tool used a multidisciplinary collaborative approach with inputs from the national IPC and port health units of the Ministry of Health and Sanitation and a WHO Country Office IPC/AMR team. This approach ensured the development of a tool that was well suited for IPC assessment at PoEs. Furthermore, the credibility and accuracy of the data were assured by having a team with the PI and staff from the National IPC and port health units. The data thus reflect the operational reality at the PoEs. Moreover, the study followed the Strengthening the Reporting of Observational Studies in Epidemiology (STROBE) guidelines in reporting [27], further improving the credibility and consistency of the data.

Our study has some limitations. First, this study did not cover all the PoEs in Sierra Leone: Low-volume crossing points (class B and Class C PoEs) were not included in this study, and comprehensive studies assessing all PoEs are needed. Second, the study was conducted during the COVID-19 pandemic, which may have influenced the results as there was an enhancement of IPC measures during this period. Third, this study has a relatively small sample size. However, the information generated in this study can serve as a baseline for further investigations. A further limitation of our study was that there was no ready-made tool for IPC assessments at PoEs. However, we believe the collaborative approach we used in adapting the WHO IPCAF tool was effective in developing a credible tool.

Despite these limitations, this study revealed important findings that can guide policy decisions. There were significant gaps in IPC guidelines, training, monitoring of practices, and isolation facilities. These should be addressed by national health authorities, following which the study should be repeated to verify that improvements have been made.

## 5. Conclusions

This was the first IPC assessment conducted at four class A PoEs in Sierra Leone, and it revealed a large number of gaps that need to be addressed. While it is relevant to Sierra Leone, it likely reflects similar situations in other low- and middle-income countries. The tool developed in Sierra Leone appeared to be feasible for and effective in identifying IPC gaps and could be adapted to other countries’ situations to improve IPC at border crossings. The findings of this study will be presented to the relevant stakeholders of the Ministry of Health and Sanitation and the One Health AMR coordination committee and partners in view of mobilizing political, financial, and material resources to address the gaps we identified.

## Figures and Tables

**Figure 1 ijerph-19-05936-f001:**
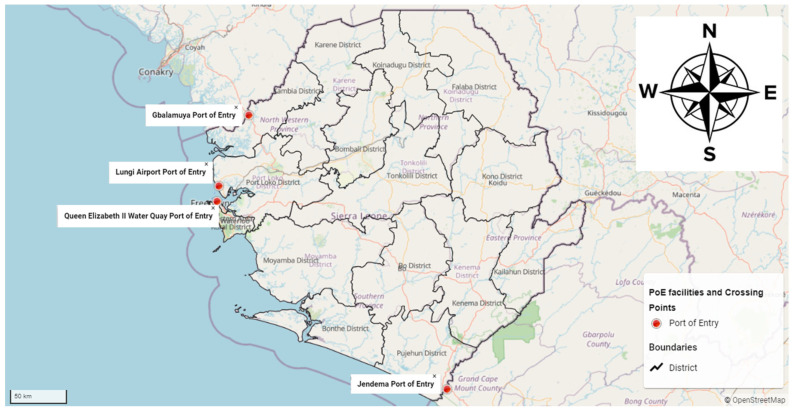
Map of Sierra Leone highlighting the four designated class A PoEs, 2021.

**Table 1 ijerph-19-05936-t001:** Levels of IPC promotion and practice for facilities from the WHO Assessment Framework (IPCAF), 2018.

Total Score (%)	IPC Level	Comments
0–25%	Inadequate	IPC core component implementation is deficient
26–50%	Basic	Some aspects of the IPC core components are in place but not sufficiently implemented
51–75%	Intermediate	Most aspects of the IPC core components are appropriately implemented
76–100%	Advanced	IPC core components are fully implemented according to WHO recommendations and appropriate to the needs of the facility

IPC = infection prevention and control.

**Table 2 ijerph-19-05936-t002:** Infection prevention and control (IPC) component scores by point of entry (PoE) type, Sierra Leone, May 2021.

	Expected Score	Queen Elizabeth II Quay	Lungi International Airport	Jendema Ground Crossing	Gbalamuya Ground Crossing
	N	N (%)	N (%)	N (%)	N (%)
**Cumulative score**	53	11 (21)	21 (40)	13 (25)	26 (49)
**Components**					
IPC guidelines	12	0 (0)	0 (0)	0 (0)	0 (0)
IPC training	6	0 (0)	3 (50)	0 (0)	4 (67)
Monitoring of IPC practices	10	0 (0)	0 (0)	0 (0)	0 (0)
Screening station	8	1 (13)	5 (63)	6 (75)	7 (88)
Cleaning and sanitation	6	3 (50)	6 (100)	1 (17)	5 (83)
Isolation facility	6	2 (33)	3 (50)	2 (33)	5 (83)
Referral system	5	5 (100)	4 (80)	4 (80)	4 (80)

**Table 3 ijerph-19-05936-t003:** Summary of IPC component scores at the four official PoEs combined, Sierra Leone, May 2021.

Core Component	Expected Score	Reported Score	Level of IPC Measures
	N	N (%)	
IPC guidelines	48	0 (0)	Inadequate
Monitoring of IPC practices	40	0 (0)	Inadequate
IPC training	24	7 (29)	Basic
Isolation facility	24	12 (50)	Basic
Screening station	32	19 (59)	Intermediate
Cleaning and sanitation	24	15 (63)	Intermediate
Referral system	20	17 (85)	Advanced

IPC = infection prevention control; PoEs = points of entry.

**Table 4 ijerph-19-05936-t004:** Identified gaps in IPC measures by PoE type, Sierra Leone, May 2021 
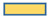
 A gap exists 
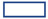
 No gap.

Guidelines	Lungi International Airport	Queen Elizabeth II Quay	Gbalamuya Ground Crossing	Jendema Ground Crossing
Guidelines on screening of travelers upon arrival or departure				
Guidelines on the isolation of sick/suspected travelers of infectious disease				
Guidelines on referral of sick or suspected passengers				
Guidelines on hand hygiene				
Guidelines on outbreak management & preparedness				
Guidelines on cleaning and disinfection				
Guidelines on waste management				
Guidelines on port health staff protection & safety				
**Training**
Port health staff received training on basic IPC				
Port cleaning staff received training on IPC				
Port administrative and managerial staff received basic training on IPC				
**Monitoring of IPC practices**
Periodic evaluations or monitoring of IPC compliance				
Trained personnel responsible for monitoring IPC practices				
Checklist to monitor IPC practices				
**Screening station**
Screening stations manned by port health staff				
Algorithm for screening available				
Screening register available				
Functional Infrared thermometer available				
Functional hand hygiene station (with soap, water and tissue)/alcohol hand rub available				
Constant water supply available for uses such as hand washing, personal hygiene and cleaning				
Adequate supply of PPE (Face mask, face shield)				
**Cleaning and sanitation**
Toilet facility available				
Adequate materials for cleaning (for example, detergent, mops, buckets, etc.) available				
Waste collection containers available				
Appropriate method of waste disposal				
**Isolation facilities**
Isolation area for suspected/sick travelers until further evaluation				
Isolation area in a permanent structure				
Isolation area standard with separate toilet and waste management				
Trained port health staff to detect travelers with suspected priority disease (e.g., COVID-19)				
PPE available at the isolation area				
Means of transportation for suspected travelers to identified healthcare facilities				

IPC = infection prevention control; PoE = point of entry; PPE = personal protective equipment.

## Data Availability

Data presented in this study are available on request from the corresponding author.

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
