# Peer review of "Assessment of Infection Prevention and Control Measures at Points of Entry in Sierra Leone in 2021: A Cross-Sectional Study"

_ijerph, 2022, doi:10.3390/ijerph19105936_

Round 1

Reviewer 1 Report

The study of the assessment of IPC measures at four major PoEs in Sierra Leone is highly innovative and important. The authors conducted a cross-sectional study using questionnaires; they concluded that the implementation levels of IPC measures at PoEs in Sierra Leone are either inadequate or basic and need to be strengthened to decrease the occurrence of AMR in the future. Overall, the manuscript is well written, and the topic is in accordance with the Special Issus: "Operational Research and Capacity Building to Tackle Antimicrobial Resistance in Sierra Leone". I only have the following minor comments for the author’s consideration.

  1. Line 19: this does not seem a correct affiliation. Please double-check.
  2. Lines 121-128: how far are these four PoEs from each other? It would be very helpful if the authors could provide a map of Sierra Leone and highlight the four designated class A PoEs.
  3. Line 192: change “principal investigator” to “PI”. It has already been abbreviated at line 189.
  4. Line 202: please indicate how many interviewed health staff from each of the four PoEs.
  5. Lines 262-264: is there any reference to support this statement?
  6. The authors appended two annexes. Please link them to the relevant content in the manuscript.

Author Response

Please find our responses to the attached document

Reviewer 2 Report

64,65, 66 things already well known, I don't think they make sense in the article

123, 138 - it's too much for this article, the description is too long

281, 282 - I don't think it has anything to do with the subject

285 -287 - the author talks about COVID screening, but not AMR

Discutions - it also requires a comparison analysis with studies conducted in other states with the same level

Conclusions - maybe there should be a recovery plan and ways to fill the gaps

Author Response

(The authors gave the same response as above.)

Reviewer 3 Report

The study provides a basic description of the implementation and monitoring of infection prevention and control measures in immigration points of entry (PoE). The manuscript is well written and technically correct. However, the study is rather a report than a scientific study. It is based on a very basic descriptive analysis of a questionnaire. The sample is very small and its representativeness questionable. 

A fundamental information that is missing relates to the size of the study and its representativeness. The authors investigate four class A PoEs with a total of 47 port health officials. The authors do not provide neither the total of port health officials in the four PoE (and the fraction of those participated) nor the corresponding individual numbers in each of the PoE. This information should be presented in detail. It may be that in some of them the numbers are so small that even a descriptive presentation is non-informative. The overall sample is very small and its validity can be partially assessed by knowing the participation fraction for the full sample and the individual PoE.

The authors emphasise that the study can contribute to the identification of Antimicrobial Drug Resistance. This is true. But the whole presentation points to the general potential of identifying infected and infectious individuals. Therefore, the emphasis on AMR is not understandable in this context. The authors may have an argument for this or weaken the introduction towards identification of infections in general.

Minor comments.

l. 88 explain abbreviation EVD

l. 89 explain class A PoE. It is explained later on but either the authors remove it here and leave the explanation further down in the manuscript or they explain it here.

line 112 explain abbreviation COVID-19

line 140 replace 'Internal' with 'International'

line 168 typo POE instead of PoE

Table 3 is two big in size with little information. It can be replaced with a simpler one with yes/no without loss of information.

line 324 supplementary material: The answer should be yes. The authors provide the questionnaire as suppl.

Author Response

(The authors gave the same response as above.)

Round 2

Reviewer 2 Report

accept in present form

Reviewer 3 Report

information in lines 205-207  belongs to section 2.7.

The sample size is too small to draw conclusions. A simple descriptive analysis than a scientific paper. No methodological novelty. Very limited information mainly descriptive. We still do not know the participation rate. The article considered 47 participants out of how many? What is the participation rate.

Author Response

Kindly see attached document
